# Prevalence and correlates of pre-diabetes in adults of mixed ethnicities in the South African population: A systematic review and meta-analysis

**Aubrey M. Sosibo**[ID]◐\*, **Nomusa C. Mzimela**‡, **Phikelelani S. Ngubane**‡, **Andile Khathi**◐

College of Health Sciences, University of Kwa-Zulu Natal, Durban, KwaZulu-Natal, South Africa

◐ These authors contributed equally to this work.
‡ NCM and PSN also contributed equally to this work.
\* SosiboA@ukzn.ac.za

## Abstract

**Data Availability Statement:** All relevant data are within the paper and its Supporting information files.

### Introduction

Pre-diabetes is a metabolic condition characterised by moderate glycaemic dysregulation and is a frontline risk factor for multiple metabolic complications such as type 2 diabetes mellitus. To the best of our knowledge, this will be the first systematic review and meta-analysis focusing on generating a comprehensive pooling of studies reporting on pre-diabetes prevalence in South Africa. Therefore, the review's purpose will be to screen and select reports that can be used to synthesise and provide the best estimate prevalence of pre-diabetes and its associated correlates in the South African population.

### Methods and analysis

To determine the prevalence and correlates of pre-diabetes in South Africa, we searched PubMed, Web of Science, Google scholar and African Journal online for published or unpublished studies reporting the prevalence of pre-diabetes in South Africa starting from the year 2000 to 2020. Studies were assessed for eligibility by checking if they met the inclusion criteria.

### Results & conclusion

The total number of studies deemed eligible is 13 and from these studies, an overall prevalence of pre-diabetes was reported to be 15,56% in the South African population. Hypertension, obesity and sedentary lifestyle were the common correlates recorded for the population of interest. Therefore, the review highlights the disturbingly high prevalence of pre-diabetes in South Africa and necessitates further investigations into the possible genetics, biochemical and hormonal changes in pre-diabetes.

**Funding:** National Research Foundation funded AM for the study. The funders had no role in study design and manuscript preparation (Grant number:121558).

**Competing interests:** The authors have declared that no competing interests exist.

## Ethics and dissemination

The review will not require ethics clearance because non-identifiable data will be used. The review outcomes will give insight into the current burden that pre-diabetes has in South Africa.

## PROSPERO registration number

CRD42020182430

## 1. Introduction

Pre-diabetes is a condition that exists when the blood glucose concentration is higher than normal but not high enough to diagnose type 2 diabetes (T2D) [1, 2]. The diagnosis of pre-diabetes can be confirmed by impaired fasting glucose (IFG), impaired glucose tolerance (IGT) or glycated haemoglobin (HbA1c) [3]. The co-existence of moderate insulin resistance and beta-cell dysfunction characterises the pre-diabetic state [2, 3]. Pre-diabetes is associated with metabolic complications often observed in T2D, such as microvascular and macrovascular complications [4–7]. Presently, lifestyle intervention and the administration of metformin are the recommended treatments for pre-diabetes [1, 8, 9].

Interestingly, the pathophysiology of pre-diabetes in subjects with impaired fasting glucose (IFG) or impaired glucose tolerance (IGT) displays different pathophysiological mechanisms [10, 11]. In people with IFG, hepatic insulin resistance is observed, while in those with IGT, muscular insulin resistance is seen [10]. In the liver cells, insulin will stimulate glycogen synthesis by activating the enzyme hexokinase, inducing the phosphorylation of glucose to glucose-6-phosphate. Simultaneously, insulin inhibits the gluconeogenesis process in the hepatic cells. However, in the IFG state, diminished insulin sensitivity in the hepatocytes will stimulate gluconeogenesis and glycogenolysis processes [11, 12]. Glucose concentration in the plasma will consequently rise.

The skeletal muscle insulin signalling pathway is initiated upon insulin binding to the α-subunit of the insulin receptor, which then causes the tyrosine residues in the intracellular β-domain of the insulin receptor to undergo autophosphorylation [13, 14]. After that, a cascade of events leads to the activation of Akt, which causes the translocation of the glucose transporter-4 (GLUT-4) to the plasma membrane, thereby allowing glucose uptake into the skeletal muscle tissue [14]. However, the insulin signalling pathway is impaired in the IGT state, resulting in glucose homeostasis dysregulation [11]. The consequences of leaving pre-diabetes unattended are significant.

Pre-diabetes is the frontline risk factor that can lead to the development of T2D [3]. This was amplified further by the American Diabetes Association (ADA) findings, which estimate that up to 70% of people with pre-diabetes will, in the long run, have T2D [15, 16]. In 2019, the global prevalence of diabetes stood at 463 million, while a further estimate of 578 million people is predicted to have T2D by 2035 [17, 18]. T2D is responsible for the deaths of approximately 1.5 million people annually [19]. In addition, other causes of fatalities, such as those related to cardiovascular disease and the more recent COVID-19 virus, occur in individuals in which diabetes exists as a comorbidity [20]. Therefore, understanding pre-diabetes can be pivotal in mitigating the incidence of T2D and the mortality rate. Consequently, this calls for further research to be conducted in investigating pre-diabetes.

The global prevalence of pre-diabetes or impaired glucose tolerance was estimated to be 7.5% (374 million) in 2019 and is projected to reach 8.0% (454 million) by 2030 [18]. Interestingly, some literature has documented the prevalence of diagnosed and undiagnosed pre-diabetes and how it affects different ethnic groups. A study by Geert Roeyen *et al.* showed that 77.7% of patients referred for pancreatic surgery had some degree of pre-diabetes [21]. Another report revealed that 15.3% of adults of the 88 million estimated to have pre-diabetes did not know, which indicates that most people remain with this condition undiagnosed [22]. This high number of undiagnosed pre-diabetes cases in people is a worrying matter because if it is left unchecked, the incidence of T2D will continue to rise [10]. Additionally, a published study conducted in Canada, a country with diverse ethnicities, has shown that different ethnic groups are disproportionately affected by type 2 diabetes in adolescents [23]. This disproportion was also evident in a study with adults conducted in England, whereby the minority ethnic groups displayed a higher prevalence of pre-diabetes [24].

Interestingly, South Africa, also known as the "rainbow nation", has a significant mixture of ethnicities, but whether certain ethnic groups in the country are disproportionally diagnosed with pre-diabetes remains unknown. Furthermore, there seems to be a paucity of pre-diabetes-related research work done within the southern African scale. Therefore, the overall prevalence of pre-diabetes in a South African population of mixed ethnicities remains uncertain or minimal.

## 1.1 Research question

1. What is the prevalence of pre-diabetes, and are any ethnic groups disproportionally affected by pre-diabetes in South Africa?

2. What are the common correlates of pre-diabetes?

## 1.2 Objectives

1. To determine the prevalence of pre-diabetes in the adult population of mixed ethnicities.

2. To determine the common correlates of pre-diabetes.

## 2. Methods

### 2.1 Design

This systemic review protocol has been prepared following the Preferred Reporting Items for Systematic Review and Meta-Analysis (PRISMA) Protocols 2015 guidelines shown in S1 Table. This review was registered in the PROSPERO International Prospective Register of systematic reviews, registration number CRD42020182430.

### 2.2 Setting

South Africa is a country on the southernmost tip of the African continent. South Africa is the largest country in Southern Africa, with over 58 million people.

### 2.3 Criteria for considering studies for the review

**2.3.1. Category of study.** The studies considered for this review consisted of cross-sectional population-based studies, prospective or retrospective cohort studies and population-

based surveys that report on the prevalence of pre-diabetes and how different ethnicities are affected. Also, a minimum of 100 participants had to be included in these studies. Furthermore, the most up-to-date and comprehensive version was selected for studies reported in more than one report.

**2.3.2. Types of participants.** The study participants included adults (≥18) located in South Africa, registered citizens who are Black, Coloured, Indian/Asian or White. The participants used in the studies had to be clinically diagnosed with pre-diabetes using the ADA or WHO diagnosis criteria.

**2.3.3. The outcome of interest.** Studies included in this review had to report the prevalence of pre-diabetes with sufficient data to calculate this estimate. Studies lacking primary data were excluded if the information was not provided after contacting authors at least twice.

**2.3.4. Diagnosis criteria.** The diagnosis criteria defined by the American diabetes association (ADA) and the World Health Organization (WHO) were considered. Accordingly, the diagnosis is determined by observing impaired fasting glucose (IFG), impaired glucose tolerance (IGT), and elevated glycosylated haemoglobin (HbA1c). IFG is defined as the fasting plasma glucose (FPG) of 6.1–6.9 mmol/L. IGT is defined as the two-hour plasma glucose of 7.8–11.0 mmol/L after the ingestion of 75 g of oral glucose load or a combination of both the IGT and the IFG recorded during the oral glucose tolerance test (OGTT). Any study that lacked a clear method description had to be excluded if, after contacting authors at least twice, the information was not provided.

## 2.4 Search strategy

Two independent (AMS & AK) reviewers conducted a comprehensive search of databases to find all related articles published on diabetes mellitus and pre-diabetes in South Africa from 2000 to 2020, regardless of the language of publication. The databases screened included MEDLINE through PubMed, Google Scholar, Web of Science and African Journals Online. The following Medical Subject Heading in our search strategy: "Pre-diabetes," "South Africa," "Prevalence," "Type 2 diabetes mellitus," "Glycated Hemoglobin," "Impaired fasting glucose," and "Impaired glucose tolerance." A detailed method is shown in the search strategy in S2 Table. The Mendeley referencing manager (V.1.19.10) was used to remove duplicates. Moreover, hand searching was done to identify other eligible studies not indexed in the databases, especially in the included studies' bibliography and relevant literature reviews.

## 2.5 Selection of included study

Two independent reviewers (AMS & AK) selected applicable studies. Each reviewer independently screened the titles, abstracts and full texts to acquire studies they consider relevant.

## 2.6 Appraisal of the quality of included studies

The methodological quality of included studies for prevalence estimates was assessed using an adapted version of the risk of bias tool for prevalence studies developed by Hoy and colleagues. A score of 0–4, 5–7, or 8–10 rated the risk of bias as high, moderate, or low, respectively. Two investigators (AMS & AK) independently assessed the study quality, with disagreements resolved by consensus or arbitration by a third review author (NM).

## 2.7 Analysis

A descriptive analysis of the eligible studies was first undertaken in the review investigation. The pooled proportion and 95% confidence interval (CI) were taken as the effect size. The

normal distribution of data was validated using the D'Agostino & Pearson omnibus normality test. After that, a Meta-analysis was undertaken using a random-effects model (to account for heterogeneity) conducted using the MetaXL (www.epigear.com) add-in for Microsoft Excel. Using MetaXL, we calculated the I2 statistics to assess heterogeneity between estimates. The I2 value describes the percentage of variation not because of chance or sampling error across studies. If the I2 value is higher than 75%, then the heterogeneity between studies is deemed high. We used the 95% CI to calculate a pooled prevalence figure.

Any possible influences on prevalence estimates were investigated using subgroup analyses. We assessed the impact on estimates of the following study-level variables identified a priori as potential sources of variation in the calculations of prevalence: [1] risk of bias, [2] geographical location, and [3] data collection method. We classified studies as being either at low risk of bias (low risk of both participation and outcome measurement bias) or moderate-to-high risk of bias (moderate or high risk of either participation or outcome measurement bias).

## 3. Results

### 3.1 Study selection

The initial search resulted in a total of 995 articles detected (Fig 1). The last search date was on the 7th of September 2021. After that, the titles and abstracts had to be screened for eligibility. Consequently, 950 articles had to be extracted due to duplication and being unrelated. Then the full-text assessment of the remaining 45 articles was conducted, which resulted in 13 articles that met the eligibility criteria.

### 3.2 Appraisal of the quality of included studies

Hoy and colleagues' risk of bias assessment for prevalence studies was used. Seven out of the 13 articles had a low risk of bias; out of the remaining seven articles, six had a moderate risk of bias, and one had a high risk of bias (Table 1). All the studies with a moderate to high risk of bias were not randomised.

### 3.3 Pre-diabetes prevalence

In a pooled sample of 5257 participants, the combined effect size for the prevalence of pre-diabetes is 15,56% in the South African population (Fig 2). The heterogeneity observed in the forest plot is 98,86%.

### 3.4 Prevalence of pre-diabetes stratified by sex and age

The results show that the female-to-male ratio is 1,18 (Fig 3). Therefore, the probability of developing pre-diabetes in females is higher by 0,18 (or 18%) than in males. In Fig 4, group A and B results show that the prevalence of pre-diabetes will generally increase with age.

### 3.5 Summary of sub-group results

The following results are seen in Table 1. The findings show that according to the geographical location, the province of Limpopo, represented by a single study, has an alarming 48,0% prevalence of pre-diabetes, which is the highest. Other provinces represented by single research include Gauteng (24,7%) and Free State (6,3%). Of the provinces that had more than one study conducted in it, the Western cape province, represented by Cape town, had the highest prevalence of pre-diabetes (15,4%), followed by KwaZulu-Natal (12,96%) and Eastern cape (9,81%). The prevalence of pre-diabetes according to race reveals that the Indian population has a proportion of 29,01%. The Indian population was followed by the Coloured at 17,38% and then

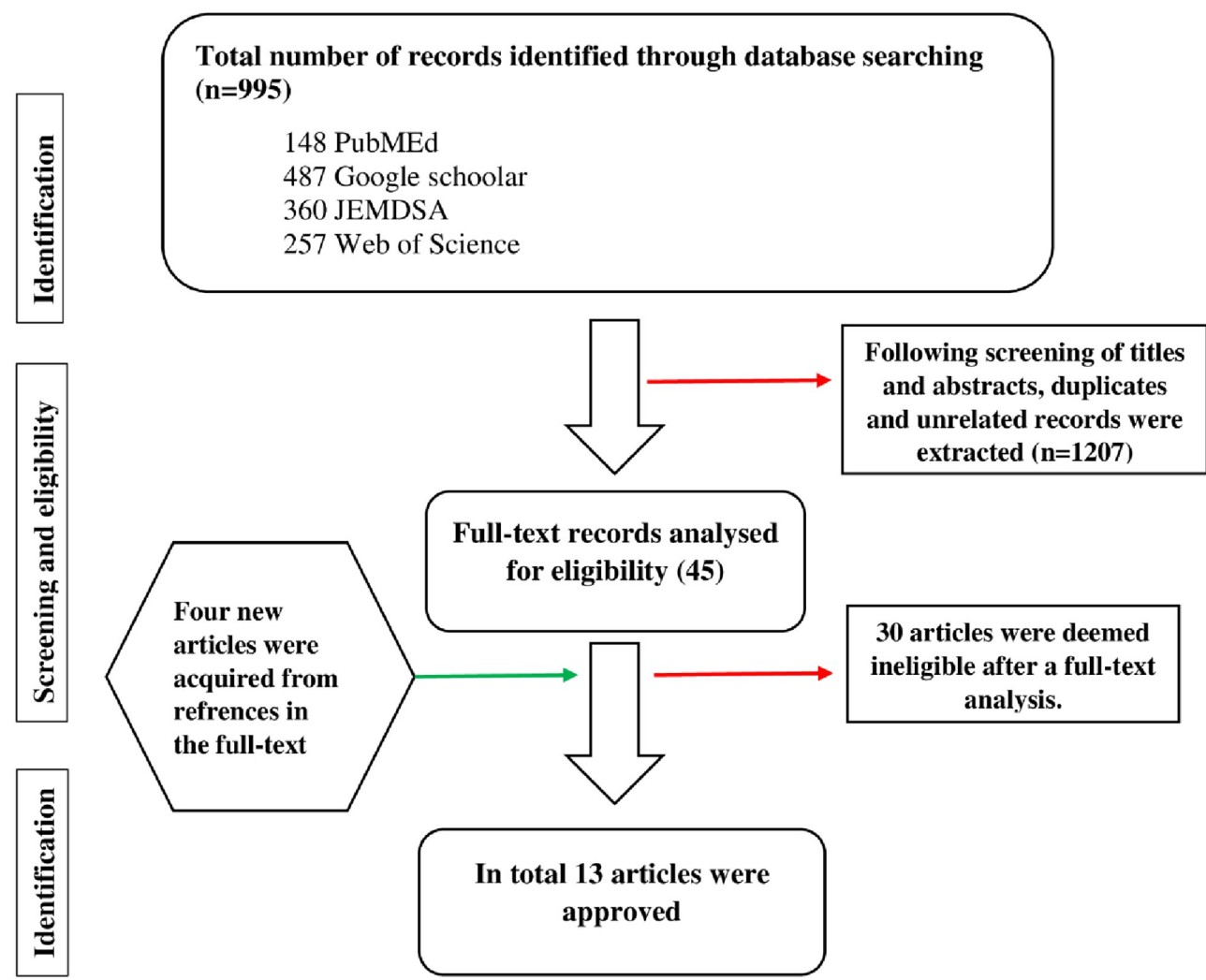

**Fig 1. Presents the procedure followed for identifying and selecting studies included in the review.**

the Black population at 13,8%. The risk of bias outcome shows that the pooled data for the group of studies with low risk is 16,7%, 15,6% for the moderate risk group and 6,1% for the high-risk group that is only represented by one study. According to the diagnosis method, the prevalence of pre-diabetes was recorded to be 18,2% in the studies that used the impaired fasting glucose test and 13,9% using the impaired glucose tolerance.

### 3.6 Correlates of pre-diabetes

The results are recorded in Table 2. From the identified correlates, hypertension has the highest total proportion of 46,8% in the pool of studies elected. People who live a sedentary lifestyle have the second highest total proportion of 43,4% in the pooled studies. In addition, the proportion of obesity (39,1%) and overweight (36,5%) is high in the population. A high number of smokers (26,3%) and people who drink alcohol (30,6%) are represented in the review. Family history of diabetes was represented by a single study that reported that 32,8% of the

**Table 1. Presents the summary of pre-diabetes prevalence in all the considered sub-groups.** Except for Limpopo, the more industrialised provinces, such as KwaZulu-Natal (KZN), Gauteng and Cape Town/Western Cape, tend to have a higher pre-diabetes prevalence. The Coloured and Indian populations have the highest prevalence of pre-diabetes. The risk of bias in most studies was low or moderate. The male and female participants have a similar prevalence of pre-diabetes. The IFG detected more pre-diabetic cases than IGT.

| Group | Studies | Participants | Cases | Prevalence % (95%CI) | I2% | p-Egger test |
|---|---|---|---|---|---|---|
| By province | | | | | | 0,009 |
| KZN | 3 | 3567 | 462 | 12,96 (-21,8–47,7) | 99 | |
| EC | 4 | 2003 | 196 | 9,81 (-0,8–20,4) | 97 | |
| CPT | 3 | 2362 | 364 | 15,42 (7,8–23,1) | 84 | |
| Free state | 1 | 552 | 35 | 6,3 | - | |
| Gauteng | 1 | 142 | 35 | 24,7 | - | |
| Limpopo | 1 | 713 | 342 | 48,0 | - | |
| By race | | | | | | 0,009 |
| Black | 10 | 6670 | 924 | 13,85 (4, - 23,6) | 99 | |
| Coloured | 2 | 1291 | 230 | 17,8 | 64 | |
| Indian | 1 | 1378 | 401 | 29,10 (26,6–31,5) | - | |
| By risk of bias | | | | | | |
| Low | 7 | 6288 | 1049 | 16,69 (1,5–31,9) | 99 | |
| Moderate | 5 | 2823 | 441 | 15,62 (7,4–23,8) | 95 | |
| High | 1 | 228 | 14 | 6,14 (3,1–9,2) | - | |
| By gender | | | | | | 0,000 |
| Male | 11 | 3066 | 586 | 19,10 (3,5–11,1) | 98,13 | |
| Female | 12 | 6472 | 1294 | 19,99 (8,8–31,2) | 98,95 | |
| By diagnosis | | | | | | 0,009 |
| IFG | 5 | 2894 | 526 | 18,19 (-3,2–39,6) | 427,94 | |
| IGT | 8 | 6445 | 895 | 13,89 (5,8–22,0) | 553,10 | |

participants had a history of diabetes. Triglyceride levels had a mean value of 1,43 mmol/L, whilst total cholesterol and LDL had mean values of 4,47 and 2,75 mmol/L.

## 4. Discussion

According to the department of Statistics of South Africa, the Republic of South Africa has a population of slightly over 60 million [25]. Therefore, the present meta-analysis showed that the overall prevalence of pre-diabetes was 15,56% in the South African population, which translates to over 9 million people with pre-diabetes. Interestingly, pre-diabetes diagnosis using the impaired fasting glucose criteria (IFG) had a higher prevalence of 18,2% compared to studies that used the impaired glucose tolerance test (IGT), which recorded 13,9%. Consequently, the heterogeneity recorded in the forest plot was 98,86%, indicating a high variability between the studies. We speculate that one reason for the high heterogeneity is the differences in the diagnosis methods utilised across the selected studies.

Differences in prevalence between IFG and IGT confirm the different pathophysiological mechanisms that contribute to these glucose homeostasis disorders. Some literature suggests that IFG is seen after impaired glucose tolerance IGT is observed [11, 26, 27]. Relative to IGT, the IFG is associated with a more significant increase in future T2DM risk [11]. Hence, IFG will generally be seen in a slightly more advanced stage of pre-diabetes prognosis relative to IGT [11]. However, compensatory mechanisms such as increased insulin secretion by the pancreatic beta-cells can delay the development of pre-diabetes [28]. Therefore, these compensatory mechanisms should be considered, as they may be more robust in specific individuals,

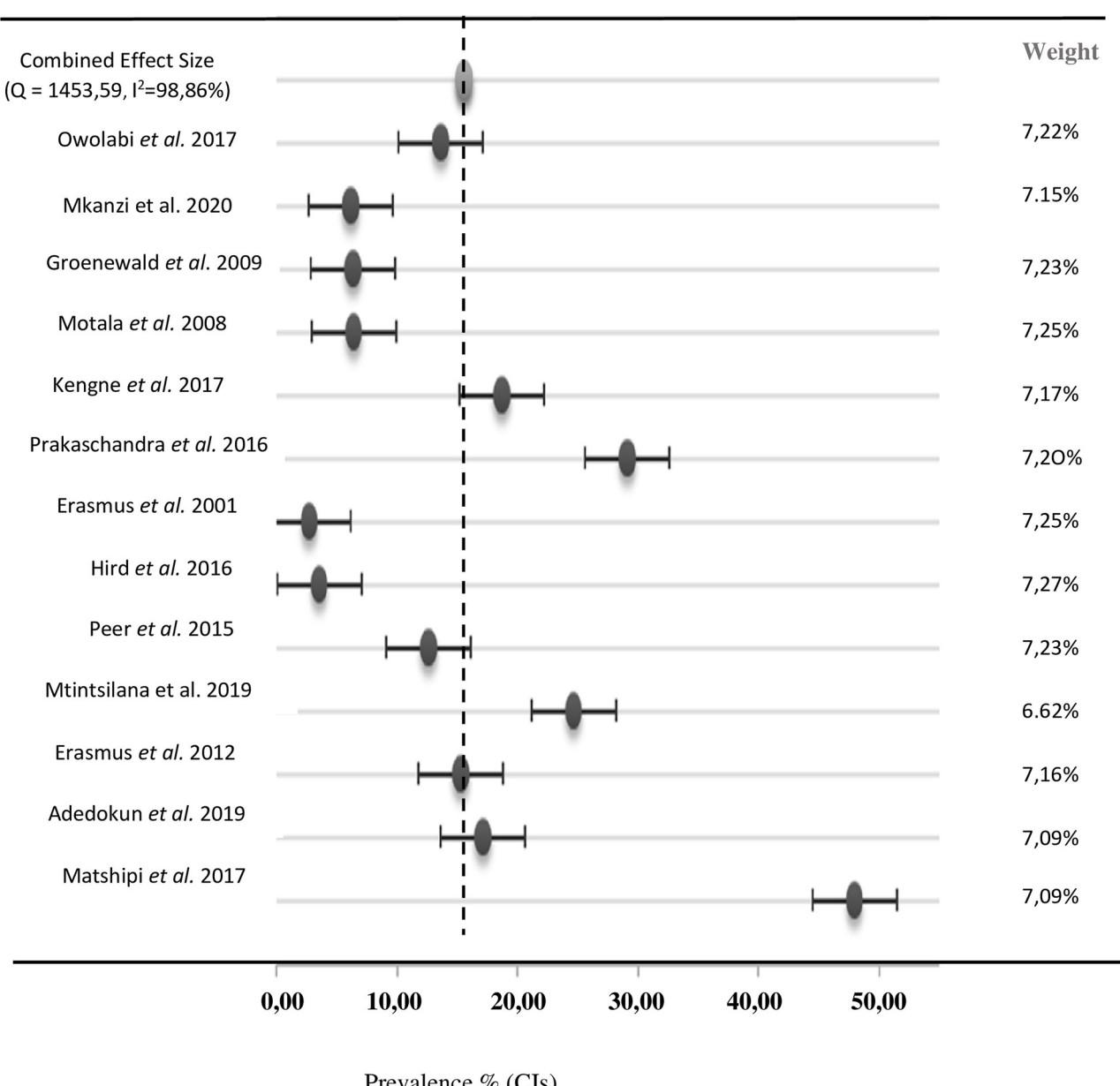

**Fig 2. Presents a forest plot of the summary of the analysis of prediabetes prevalence in South Africa.** The diagnosis was determined by either IFG or IGT test. In a pooled sample of 9339 participants, the prevalence of prediabetes was 15,56%, with a heterogeneity of 98,86%.

allowing them to bypass the early phase of IGT. Hence, it is plausible that at this moment, a speculate that a more significant number of people with pre-diabetes remain undiagnosed either because they do not go for health screening or because the clinicians may prefer to use only IFG to determine pre-diabetes. By doing so, they miss many pre-diabetic individuals according to the IGT criteria.

The sex of the participant seems to play a slight role in developing pre-diabetes. The results in Fig 2 indicate that females have an 18% greater chance of developing pre-diabetes when compared to males. This contrasts with what some literature suggests: that males are more

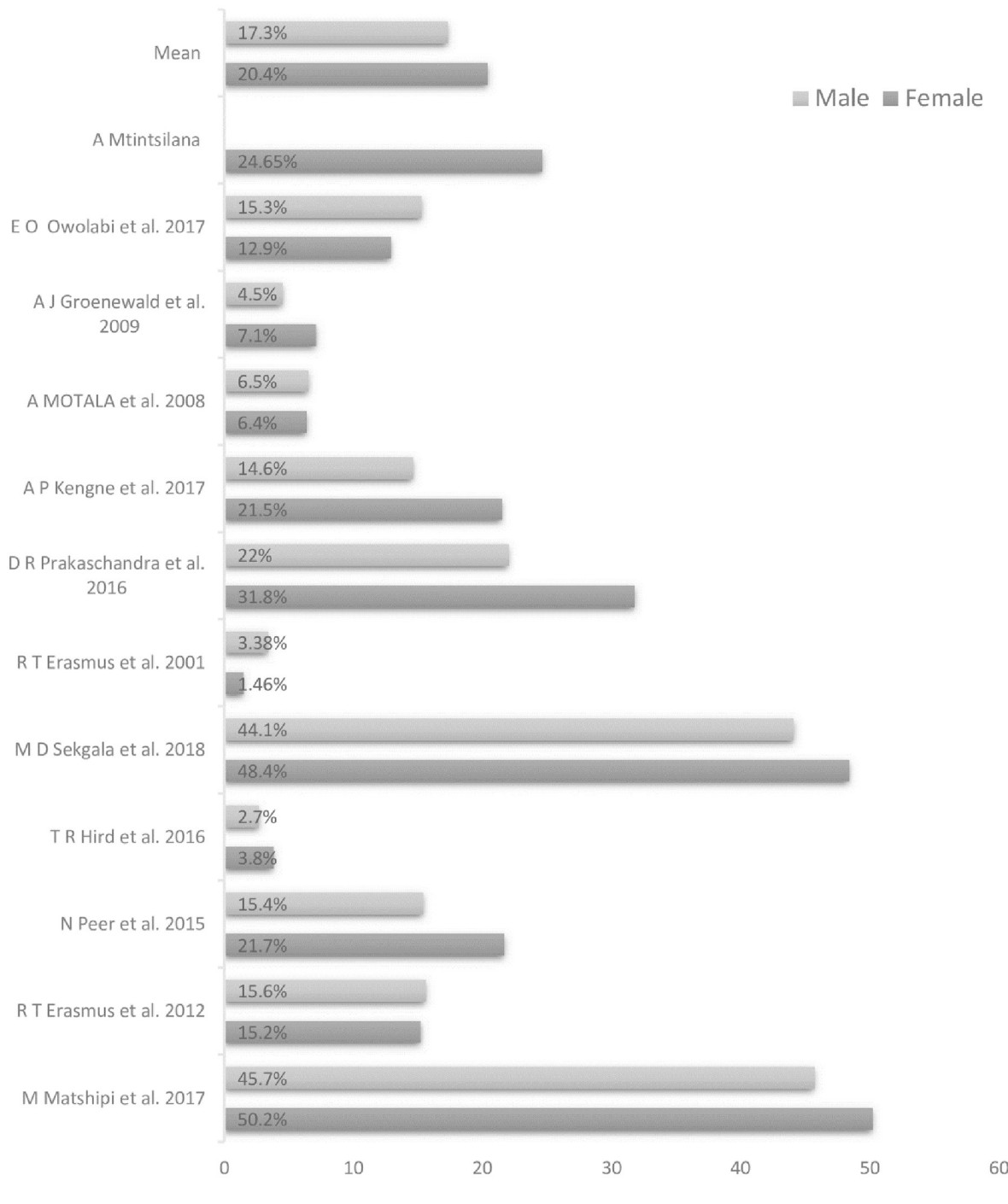

**Fig 3. Presents the prevalence of pre-diabetes stratified by sex in the South African population.** The study conducted by A Mtintsilana only had female participants. The overall female-to-male ratio is 1,18.

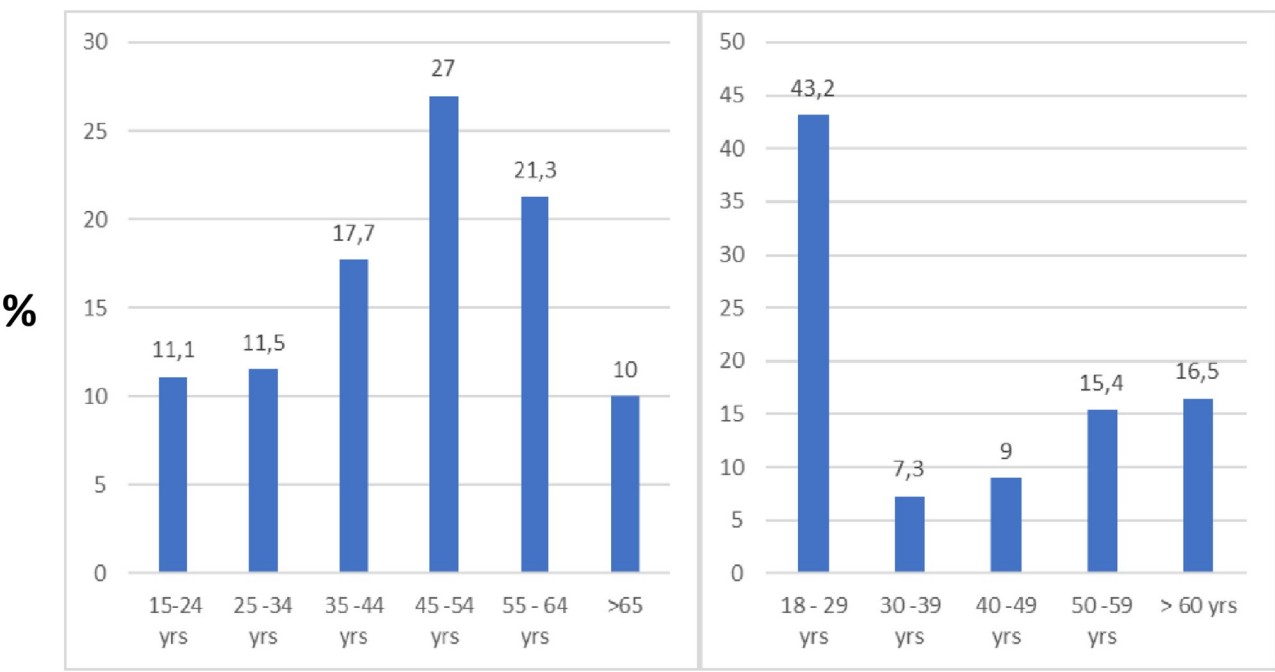

Group A: Age starting from 15

Group B: Age starting from 18

**Fig 4. Presents the prevalence of pre-diabetes stratified by age.** Except for the 18–29 age range in group B, the proportion of individuals with prediabetes tends to increase with increasing age.

susceptible to diabetes than females [29–31]. Their justification is that men tend to store fat in their organs, like the liver, compared to women, who store fat mainly on the surface layer, such as the hip and thighs [31]. However, literature has also shown that females are more prone to developing IFG than males, independent of age [32, 33]. This suggests that females

**Table 2. Shows the proportion of risk factors associated with pre-diabetes.**

| Studies | obesity | overweight | hypertension | tobacco | alcohol | Triglycerides | Chol (mmol/l): Total & LDL | Family history | Sedentary lifestyle |
|---|---|---|---|---|---|---|---|---|---|
| Matshipi *et al.*, 2017 | - | - | - | - | - | - | - | - | - |
| Adedokun *et al.*, 2019 | - | - | - | | 49.4 | - | - | - | - |
| Erasmus *et al.*, 2012 | - | - | - | 44.4 | 28.4 | - | - | - | - |
| Mtintsilana *et al.*, 2019 | 70.4 | 21.1 | - | - | - | - | - | - | - |
| Peer *et al.*, 2015 | 63 | 58 | 49 | 28 | 33 | - | 4.4 & 3.0 | - | - |
| Hird *et al.*, 2016 | 37.7 | 60.8 | 28.9 | 19.2 | 14.3 | 1.4 | 4.3 & 2.3 | 32.8 | 46.5% |
| Erasmus *et al.*, 2001 | 10 | 10 | - | - | - | - | - | - | - |
| Prakaschandra *et al.*, 2016 | 10,4 | - | 79 | 24,9 | - | 1.8 | 3.35 | - | - |
| Kengne *et al.*, 2017 | - | - | - | - | - | - | 3.3 & 5.2 | - | - |
| Motala *et al.*, 2008 | - | 64 | 64 | - | - | 1.1 | 4.1 & 2.4 | - | - |
| Groenewald *et al.*, 2009 | 32.1 | 20.8 | - | - | - | - | - | - | 9.1% |
| Mkanzi *et al.*, 2020 | 44 | 33 | 11 | - | - | - | - | - | - |
| Owolabi *et al.*, 2017 | 45.2 | 24 | 49 | 15 | 28 | - | - | - | 74.7% |
| **Mean** | 39,1 | 36,5 | 46,8 | 26,3 | 30,62 | 1,43 | 4,27 & 2,75 | 32,8 | 43.4 |

may have more robust protective mechanisms that delay the onset of T2D. A European study corroborated that men are often diagnosed with T2D earlier than women [32, 34]. Therefore, the higher prevalence of pre-diabetes observed in women can be explained by their greater susceptibility to developing pre-diabetes.

Another interesting factor that the review looked at is age. The findings in this review show that the association between age and the prevalence of pre-diabetes are directly proportional. The results are consistent with the literature as the prevalence of pre-diabetes rises with increasing age [24, 35, 36]. The plausible explanation for the correlation between age and prevalence is that people tend to live a more sedentary lifestyle as they grow older. A study by Zoran Milanović *et al*. revealed that relative to the young elderly, there is a reduction in physical activity amongst the old elderly [37]. The decline in physical activity was attributed to decreased muscle strength in both upper and lower limbs and changes in body-fat percentage [37]. However, the high prevalence observed in the 18–29 age group in Fig 4 can be attributed to the study by Sekgala et al., which reported on the 18–29 age group only [38]. Therefore, it is plausible to expect a similar trend had the study by Sekgala included the older age groups.

The prevalence of pre-diabetes according to geographical location shows that the Limpopo and Gauteng provinces have the highest prevalence of 48% and 24,8%, respectively. However, this cannot be taken as conclusive data because these provinces have a small sample size and are represented by a single study. Therefore, more prevalence studies are required in these provinces. The outcomes of the prevalence of pre-diabetes according to the geographical location are more credible in the provinces with multiple studies. These provinces include the province of the Western Cape, represented by Cape Town, which had the highest prevalence of 15,42%, followed by KwaZulu-Natal with 12,96% and Eastern cape with 9,81%. However, ethnicity is known to play a role in the prevalence of diabetes.

The prevalence of pre-diabetes across different ethnic groups in South Africa suggests a genetic predisposition for developing pre-diabetes. The Indian population have the highest prevalence of 29,01%, followed by the coloured community with 17,38% and the black population with 13,8%. Therefore, ethnicity plays a significant role in predisposition to developing pre-diabetes. There is increasing evidence in the literature on how epigenetic factors play an essential role in the development of T2D [39, 40]. Epigenetics can be described as transmissible alterations in gene functions that arise with no changes in their nucleotide sequence [41]. Epigenetic markers like miRNAs and methylation are shown to be involved in the pathogenesis of diabetes and may explain why some ethnic groups are disproportionally affected by pre-diabetes [23, 40, 42]. Hence, further studies appraising the role of epigenetics in developing pre-diabetes are needed for the population of South Africa. Furthermore, pre-diabetes is associated with numerous other medical conditions.

Unsurprisingly, the prevalence of pre-diabetes correlates with high occurrences of hypertension, obesity, overweight, and a sedentary lifestyle. These risk factors are well documented in the literature to be associated with the development of diabetes [24, 43, 44]. Hence, there is expected to be a high pre-diabetes prevalence whenever these correlates are high in a population. Therefore, we must not look at pre-diabetes as an isolated condition; instead, we need to monitor its risk factors, which play a significant role in the onset and prognosis of pre-diabetes. Derek Weycker *et al*. have shown that regardless of BMI, age, and sex, all people with hypertension are at a considerable risk of becoming diabetic [45]. Also, the sedentary lifestyle over the years has risen with the increasing urbanisation and consumption of western fast foods, which according to the literature, is directly proportional to the rise in pre-diabetes prevalence and all course mortality [46, 47]. A sedentary lifestyle causes an imbalance in calorie intake, resulting in a greater intake than output. This imbalance causes the body to store the extra energy source as fats which accumulate and cause insulin resistance that leads to pre-diabetes

and type 2 diabetes [48, 49]. Likewise, the obesity and overweight association with diabetes is well established [50]. Both conditions of obesity indicate early signs of insulin resistance and can result in the development of diabetes if unattended. Therefore, when assessing the prevalence of pre-diabetes in a particular location, these associated risk factors must also be monitored to identify those at risk.

## 5. Limitations

One of the limitations of this study is that only seven of the 13 eligible studies had a low level of risk. This suggests that more credible studies are needed to have a pool of studies that will generate a more reliable overall prevalence in the country. Furthermore, some utilised the outdated version of the WHO diagnostic criteria for impaired glucose tolerance and impaired fasting glucose. The lack of a universal definition for determining pre-diabetes is also a limitation for this review because different studies utilised different criteria for deciding pre-diabetes.

## 6. Conclusion

The selected studies did not apply the same criteria for diagnosing pre-diabetes and some used outdated measures. Consequently, there is a need for a universal diagnostic marker for pre-diabetes that will improve accuracy when comparing and pooling epidemiological studies. Furthermore, the present review findings are concerning because a high prevalence of pre-diabetes poses a significant health care burden. Pre-diabetes is associated with metabolic disorders and is the frontal risk factor for developing the more harmful type 2 diabetes. Thus, there is a gap for more prevalence studies on prediabetes that will use current diagnostic criteria for pre-diabetes and reveal updated data on the prevalence of pre-diabetes.

Additionally, as seen in T2D, there appears to be a correlation between a high prevalence of pre-diabetes and hypertension, obesity and a sedentary lifestyle. Screenings done in the hospitals should therefore be holistic by including markers for pre-diabetes and associated correlates. Finally, the findings suggest that clinicians should put in place preventative measures for preventing the progression from pre-diabetes to type 2 diabetes and have more clinical outreaches to identify early those who have pre-diabetes.

## Supporting information

**S1 Table. PRISMA-P 2015 checklist.**
(DOCX)

**S2 Table. Search strategy (PubMed).**
(DOCX)

## Acknowledgments

The authors would like to acknowledge everyone in the Endorince research group for their academic and moral support.

## Author Contributions

**Conceptualization:** Aubrey M. Sosibo, Andile Khathi.

**Data curation:** Aubrey M. Sosibo.

**Formal analysis:** Aubrey M. Sosibo, Phikelelani S. Ngubane, Andile Khathi.

**Funding acquisition:** Aubrey M. Sosibo, Andile Khathi.

**Methodology:** Aubrey M. Sosibo, Nomusa C. Mzimela.

**Project administration:** Aubrey M. Sosibo.

**Supervision:** Phikelelani S. Ngubane, Andile Khathi.

**Validation:** Nomusa C. Mzimela, Andile Khathi.

**Writing – original draft:** Aubrey M. Sosibo.

**Writing – review & editing:** Aubrey M. Sosibo, Nomusa C. Mzimela, Phikelelani S. Ngubane, Andile Khathi.

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
