## [Decision Letter · Decision Letter 0]

4 Aug 2022

PONE-D-22-14937Prevalence and correlates of pre-diabetes in adults of mixed ethnicities in the South African population: A Systematic Review and Meta-analysis

PLOS ONE

Dear Dr. Sosibo,

Thank you for submitting your manuscript to PLOS ONE. After careful consideration, we feel that it has merit but does not fully meet PLOS ONE’s publication criteria as it currently stands. Therefore, we invite you to submit a revised version of the manuscript that addresses the points raised during the review process.

I have received the reports from our advisors on your manuscript which you submitted to PLOS ONE.

Based on the comments received, I feel that your manuscript could be reconsidered for publication should you be prepared to incorporate major revisions.

When preparing your revised manuscript, you are asked to carefully consider the reviewer comments below and submit a list of responses to the comments.

Editor Comments: There is a huge list of grammatical mistakes and syntax errors. The paper should be checked by a professional speaker of English before complete acceptance.

We look forward to receiving your revised manuscript.

Kind regards,

Muhammad Sajid Hamid Akash

Academic Editor

PLOS ONE

Journal Requirements:

   "I would like to acknowledge the National Research Foundation for funding the study (Grant number:121558) and everyone in the Endorince research group fortheir academic and technical support."

  "National Research Foundation funded AM for the study. The funders had no role in study design and manuscript preparation (Grant number:121558)"

3. We note you have included a table to which you do not refer in the text of your manuscript. Please ensure that you refer to Table 2 in your text; if accepted, production will need this reference to link the reader to the Table.

Reviewers' comments:

Reviewer's Responses to Questions

**Comments to the Author**

1. Is the manuscript technically sound, and do the data support the conclusions?

Reviewer #1: Yes

2. Has the statistical analysis been performed appropriately and rigorously? 

Reviewer #1: Yes

3. Have the authors made all data underlying the findings in their manuscript fully available?

Reviewer #1: Yes

4. Is the manuscript presented in an intelligible fashion and written in standard English?

Reviewer #1: Yes

5. Review Comments to the Author

Reviewer #1: The author presented an original article on “Prevalence and correlates of pre-diabetes in adults of mixed ethnicities in the South African population: A Systematic Review and Meta-analysis”. The manuscript needs to be improved regarding the following aspects.

• More details are required to explain the association between key points and the conclusion.

• Comprehensively explain the disease pathogenesis, its associated complications and the therapeutic interventions to strengthen your study e.g., https://doi.org/10.1002/jcb.26174 , https://doi.org/10.1111/jfbc.14228 and https://doi.org/10.1615/CritRevEukaryotGeneExpr.2015013358 ,

• Results are not well explained and comprehensively described.

• More graphical representations of the results should be included in the manuscript.

• More description is required in the discussion section to elaborate the association between the results and the literature review.

• Many references are missing or inconsistent in format.

• The format of the manuscript is inconsistent and is required per the requirement of the journal.

• There are some grammatical mistakes in the manuscript, such as verbs and prepositions. The manuscript needs extensive review by an author.

6. PLOS authors have the option to publish the peer review history of their article (what does this mean?). If published, this will include your full peer review and any attached files.

Reviewer #1: **Yes: **Sumbal Rasheed

---

## [Author Response · Author response to Decision Letter 0]

12 Sep 2022

The authors would like to thank the editor and reviewer for their comments that helped improved the manuscript. All the comments were thoroughly attended to as seen in the uploaded rebuttal letter and highlighted manuscript.

---

## [Decision Letter · Decision Letter 1]

15 Nov 2022

Prevalence and correlates of pre-diabetes in adults of mixed ethnicities in the South African population: A Systematic Review and Meta-analysis

PONE-D-22-14937R1

Dear Dr. Aubrey Mbulelo Sosibo,

We’re pleased to inform you that your manuscript has been judged scientifically suitable for publication and will be formally accepted for publication once it meets all outstanding technical requirements.

Kind regards,

Paolo Magni

Academic Editor

PLOS ONE

Additional Editor Comments (optional):

All comments have been addressed

Reviewers' comments:

Reviewer's Responses to Questions

**Comments to the Author**

1. If the authors have adequately addressed your comments raised in a previous round of review and you feel that this manuscript is now acceptable for publication, you may indicate that here to bypass the “Comments to the Author” section, enter your conflict of interest statement in the “Confidential to Editor” section, and submit your "Accept" recommendation.

Reviewer #1: All comments have been addressed

2. Is the manuscript technically sound, and do the data support the conclusions?

Reviewer #1: Yes

3. Has the statistical analysis been performed appropriately and rigorously? 

Reviewer #1: Yes

4. Have the authors made all data underlying the findings in their manuscript fully available?

Reviewer #1: Yes

5. Is the manuscript presented in an intelligible fashion and written in standard English?

Reviewer #1: Yes

6. Review Comments to the Author

Reviewer #1: (No Response)

7. PLOS authors have the option to publish the peer review history of their article (what does this mean?). If published, this will include your full peer review and any attached files.

Reviewer #1: **Yes: **Sumbal Rasheed

---

## [Editor Report · Acceptance letter]

17 Nov 2022

PONE-D-22-14937R1 

Prevalence and correlates of pre-diabetes in adults of mixed ethnicities in the South African population: A Systematic Review and Meta-analysis 

Dear Dr. Sosibo:

I'm pleased to inform you that your manuscript has been deemed suitable for publication in PLOS ONE. Congratulations! Your manuscript is now with our production department. 

Kind regards, 

on behalf of

Prof. Paolo Magni 

Academic Editor

PLOS ONE